# The European Ground Squirrel's Genetic Diversity in Its Ancestral Land: Landscape Insights and Conservation Implications

Yordan Koshev [1,*], Štěpánka Říčanová [2,3], Maria Kachamakova [1] and Oldřich Říčan [2]

1   Institute of Biodiversity and Ecosystem Research, Bulgarian Academy of Sciences, 1 Tzar Osvoboditel Blvd, 1000 Sofia, Bulgaria
2   Department of Zoology, Faculty of Science, University of South Bohemia, Branišovská 31, 370 05 České Budějovice, Czech Republic
3   Southbohemian Zoo Hluboká nad Vltavou, Ohrada 417, 373 41 Hluboká nad Vltavou, Czech Republic
*   Correspondence: bgsouslik@gmail.com

**Abstract:** The European ground squirrel is an endangered rodent whose populations are declining throughout its range. Only in Bulgaria, the genetic hotspot of the species, are some abundant populations still present. We employed 12 microsatellite loci in ten Bulgarian populations to look at population structure, gene flow and recent bottlenecks. We found that the populations are in good condition in terms of heterozygosity, where values ranged from 0.55 to 0.78. However, the inbreeding index ($F_{IS}$) was significant for most populations. A recent bottleneck was detected in only one population. Based on Bayesian clustering methods, the populations in Bulgaria were attributed to two groups, northern and southern, with admixture in the northern one. The AMOVA test between these groups showed no differentiation in genetic diversity. The mean value of $F_{ST}$ was 0.184, which shows strong diversification among all populations. Hence, gene flow is probably limited. All these results indicate that Bulgaria is the main area to focus the efforts for conservation of the species by ensuring that the complex and rich genetic structure of Bulgarian populations is preserved.

**Keywords:** population genetics; microsatellite; *Spermophilus citellus*; Sciuridae; souslik; Bulgaria





## 1. Introduction

Much of the landscape of our planet has been transformed into monoculture croplands to support the 8 billion human beings alive today. The replacement of wilderness by human exploited environments is causing a rapid loss of biodiversity at species and ecosystem levels throughout the world [1–3]. In some regions, such as in Europe, many species have to face a human-altered environment and even some rodent species, usually considered to be very flexible and opportunistic, have been affected by increasing alteration of their natural environment [4–6]. Genetic diversity is commonly considered as a fundamental key factor for the long-term viability of populations and is crucial for conservation. The data on the genetic structure of populations help in identifying hotspots of genetic diversity—geographic areas harbouring a major portion of a species' genetic diversity [7]. Further, landscape connectivity is also very important in the sense of the viability and prosperity of species [8–11]. Landscape genetics is a multidisciplinary science, combining population genetics, molecular ecology, biogeography, evolutionary biology and systematics [12]. Nowadays, a well-defined approach for studying the interactions between landscape features and evolutionary processes, mainly gene flow and selection in nature and urban populations, has been established, and novel methods and technology have been introduced (reviewed in [13]). To understand the landscape features that limit gene flow, the spatial scales at which they act and the temporal dynamics of their effects on population substructure, it is essential to effectively use genetic data as a tool

for evaluating population status and fragmentation. Using the accumulated knowledge to predict, localise and implement empirically based ecological corridors could greatly improve the successfulness of efforts to promote landscape connectivity of species at risk due to fragmentation [14]. In addition, the effect of a population bottleneck is directly related to the increase in stochastic events associated with small population size, which in most cases leads to further loss of genetic diversity [15]. Natural immigration and rescue effects (human-mediated) for bottlenecked populations could increase genetic variability and reduce inbreeding. On the other hand, negative genetic effects are not always present. For example, in the golden-mantled ground squirrel no evidence of increased inbreeding during or after the decline was recorded [16], and the major factors of local population extinction risk in this species are more likely to be demographic (reproduction, immigration, predation) than genetic in origin. Despite these exceptions, it is generally believed that sufficient levels of immigration and gene flow within the regional meta-population are critical for the long-term survival of populations and highlight the importance of maintaining connectivity in natural populations [17,18].

In this study, we focus on the European ground squirrel (EGS) (*Spermophilus citellus* Linnaeus, 1766). It is an obligate hibernator and typically inhabits steppes, meadows and pastures from sea level to 2500 m a.s.l. Distributed in central Europe and the eastern Balkans, it is the westernmost member of the genus *Spermophilus* [19]. Bulgaria is situated in the southern part of its range and has a diverse topography with many different habitats (mosaic of forests, pastures, meadows and arable lands). In some mountains the species is found in open habitats surrounded by vast forested areas [20,21]. Based on a previous study [22], Bulgaria hosts the most viable populations and represents the ancestral area of the species, since the main phylogenetic lineages occur there. Both the Northern and Southern lineages also showed the highest genetic diversity in Bulgaria. In the past, the species' range was characterised by the glacial and interglacial dynamics of the steppe habitat. Today, the range is progressively shrinking and the species is considered globally endangered [23], with declining numbers especially at its western and southernmost margin [24,25]. In Bulgaria during the last several decades, about 30% of investigated colonies have disappeared, 28% are vulnerable and only 42% are stable [20]. Major threats are pasture degradation, building, intensification of agriculture, interruption of biological corridors and flooding. The ground squirrel population in Bulgaria probably continues to decline and its current species status is unfavourable [20,21]. Additionally, the species is still poorly protected by Bulgarian laws. Only its habitats have some low level of protection under the Biodiversity Act and through the Natura 2000 network of protected sites [20,21]. The EGS is included in the Red Book of Animals in Bulgaria as "Vulnerable" [26].

The aims of this study were to investigate the genetic diversity and population structure of *S. citellus* in Bulgaria using microsatellite markers, as well as to suggest management recommendations for conservation of the species based on the results. Taking into account the results of other phylogenetic studies on EGS [27–29] and other related species, we hypothesised that mountain chains and big rivers would play barrier role for the genetic flow together with the recent socioeconomic processes, thus shaping the patterns of genetic diversity and relatedness among the Bulgarian populations.

## 2. Material and Methods

### 2.1. Study Area, Sampling and Dataset

This study is based on 173 tissue samples of the EGS from 10 populations in Bulgaria: NP (Nikopol, Pleven district); TS (Tsenovo, Ruse district); BN (Belmeken, Rila Mountains, Pazardzhik district); KZ (Knezha, Pleven district); KAP (Kap. Petko Voivoda, Haskovo district); TOP (Topolchane, Sliven district); RZ (Rozino, Plovdiv district); KRE (Kremikovtsi, Sofia district); CG (Chernogorovo, Pazardzhik district); and ISH (Professor Ishirkovo, Silistra district) (see Figure 1). The individuals were trapped during their active season using live traps. Sampled individuals were caught evenly across each locality (colony) to minimise the sampling of close relatives. From each individual a small piece of ear was cut

off and stored in pure 96% ethanol until DNA extraction. The size of population samples varies from 6 to 25 individuals per population (mean ± SD: 17.4 ± 5.96); more details are found in Table S1, see Supplementary Materials.

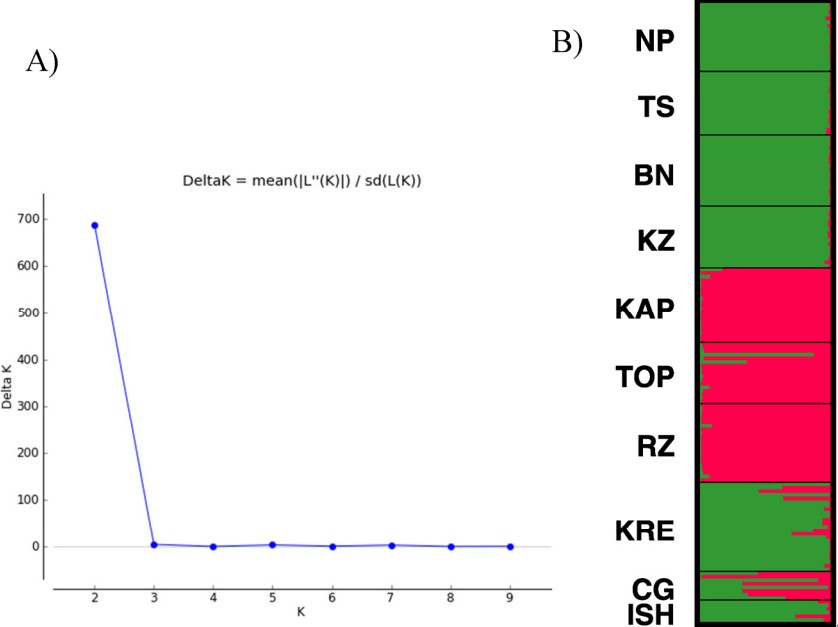

**Figure 1.** Bayesian clustering from Structure. (**A**) The best probabilities for K according to Evanno et al. (2005) using ΔK. (**B**) Population assignment to the two clusters, i.e., green—northern Bulgarian, red—southern Bulgarian populations. Each individual is represented by a horizontal bar and the whole sample is divided into K colour-coded segments. Bold black lines separate predefined input populations.

The microsatellite dataset was obtained in the study of Říčanová et al. [22]. See there for methodology. Here, we analysed the dataset in a different context, with a focus on Bulgaria rather than on a whole species context and with new insights on the time scale and a finer scale interpretation of the landscape features.

### 2.2. Intra-Population Analysis, Spatial Genetic Structure and Gene Flow

Allelic richness, observed (*Ho*) and expected (*He*) heterozygosity, inbreeding coefficient ($F_{IS}$) values and linkage equilibrium for each pair of loci were calculated using FSTAT v. 2.9.3 [30]. Test of significance in $F_{IS}$ was tested in Genetix v.4.05 [31]. Global Hardy–Weinberg tests were run in FSTAT to assess deviations from Hardy–Weinberg equilibrium and a significance level of linkage disequilibrium for each pair of loci. We employed FreeNA software [32] to estimate null allele frequencies for each locus and population following the expectation maximization algorithm of Dempster et al. [33]

BOTTLENECK v. 1.2.02 [34] was used to investigate a possible recent reduction in effective population size. Genetic data were analysed under the two-phase mutation model, with 90% of the mutations following a stepwise mutation model and variance of 30%. Wilcoxon sign-rank test was used for the test of significance [35]. Population structure was estimated as pairwise and overall values of $F_{ST}$ (fixation index). FreeNA software [32] was used to estimate unbiased $F_{ST}$ [36] following the ENA (excluding null alleles) correction method described in Chapuis and Estoup [32]. Isolation by distance (IBD) was tested by performing Mantel tests [37], i.e., correlation between $F_{ST}/(1 - F_{ST})$ and the logarithm of geographic distances using Arlequin 3.1 with 100,000 permutations between pairs of populations [38]. We used Slatkin's linearized $F_{ST}$'s.

Further, two Bayesian methods implemented in the programs Structure v. 2. [39,40] and BAPS 5.1. [41] were used to investigate the geographical distribution of genetic variabil-

ity. Markov chain Monte Carlo (MCMC) algorithms were employed to group individuals (for Structure) or populations (for BAPS) in clusters (the number of clusters is a priori unknown) to achieve Hardy–Weinberg equilibrium and linkage equilibrium within each cluster. The program BAPS was run 15 times with a burn-in period of 10,000 iterations followed by 50,000 iterations. Neighbour-joining (NJ) tree was drawn based on Nei genetic distances to see the clusters and try to find a geographical pattern. Further, another 15 runs were run with fixed K in BAPS. The program Structure was run 10 times for each K value from 1 to 10, each run comprising a burn-in period of 1,000,000 iterations followed by 100,000 iterations. In Structure, the admixture model and the independent allele frequencies model were used [39,42]. Δ K, an ad hoc statistic based on the second order rate of change of the likelihood function with respect to K, was used and evaluated as the most likely number of populations [43]. We used a Structure Harvester program [44] to determine the number of genetic clusters from Structure analyses in order to compare the mean likelihood and variance per K values based on the 10 independent runs. The Greedy algorithm of Clumpp 1.1.1 [45] was used to combine the results from 10 runs for each K and the summary results were graphically visualized by Distruct v.1.1 [46]. Additionally, an AMOVA test [47] implemented in Arlequin 3.1. [38] was used to estimate the proportion of genetic variation explained by the estimated K clusters.

## 3. Results

### 3.1. Genetic Diversity

The allelic richness per population ranged from 3.34 (BN) to 5.30 (TOP). The overall genetic diversity *He* ranged from 0.555 (TS) to 0.784 (TOP). The test for HWE showed strong deviation from HWE only for the population of KRE. The results suggest the presence of null alleles in microsatellites loci, the frequency of which was estimated with FreeNA, to vary from 0.024 to 0.142. All genetic indices are summarised in Table 1. No linkage disequilibrium was recorded in the studied loci.

**Table 1.** Comparison of genetic diversity in N (northern Bulgarian) and S (southern Bulgarian) populations in allelic richness (AR), observed (*Ho*) and expected (*He*) heterozygosity, coefficient of inbreeding ($F_{IS}$), genetic differentiation ($F_{ST}$) and corrected relatedness coefficient (Rel). Level of significance is considered $p < 0.05$.

|  | **N** | **S** | ***p*** |
|---|---|---|---|
| AR | 3.927 | 4.597 | 0.12880 |
| *Ho* | 0.541 | 0.587 | 0.32040 |
| *He* | 0.646 | 0.722 | 0.21167 |
| $F_{IS}$ | 0.162 | 0.187 | 0.83180 |
| $F_{ST}$ | 0.179 | 0.174 | 0.89720 |
| Rel | 0.273 | 0.262 | 0.87313 |

Wilcoxon tests of recent bottlenecks revealed that only one of the populations showed significant departures from mutation-drift equilibrium, i.e., the population of NP ($p = 0.007$) (Table 1).

### 3.2. Spatial Clustering

Two Bayesian clustering softwares were used to assess population structure and to assign individuals to populations according to their genotypes, i.e., Structure v. 2 and BAPS v. 5.1. Both approaches brought almost the same results. In Structure, when we performed a method for the estimation of K according to Evanno et al. [43], using Δ K as the best estimation for the most likely number of populations, the populations were divided into two groups (Figure 2): southern Bulgarian group (KAP, TOP and RZ) and northern Bulgarian group (NP, BN, KZ and TS) (Figures 1 and 2). Populations KRE, CG and ISH were assigned as mixtures of these groups.

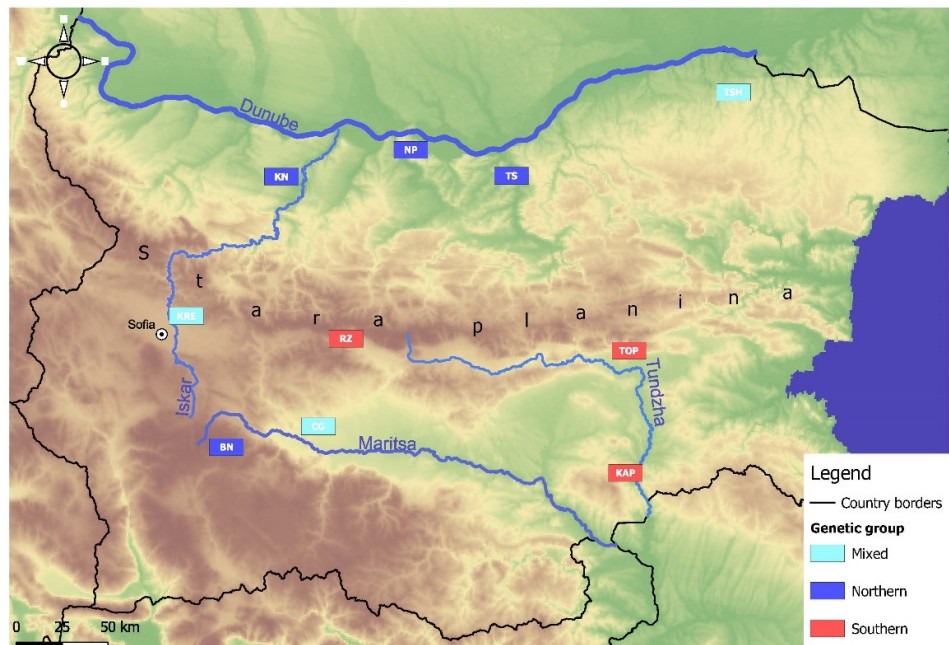

**Figure 2.** The sampled locations presented on the topographic map of Bulgaria. The different genetic groups based on Structure results (Figure 1) are shown with different colours according to the legend. The altitude is presented with gradual colouring from green (low altitude from 0 m a.s.l.) to brown (mountains up to 2925 m a.s.l.).

In BAPS, it was found out that each population represents an individual unit; hence, the populations were divided into 10 groups (Figure 3A). When a NJ tree was performed based on Nei distances, these 10 groups were clustered into two groups as well (Figure S1, Supplementary Materials). The first group of populations consisted of ISH, TS, NP, KZ, KRE and BN, i.e., the northern Bulgarian group of populations. The second group included RZ, CG, TOP and KAP, i.e., the southern Bulgarian group (Figure 3B). The two clusters were thus identical to those identified in Structure.

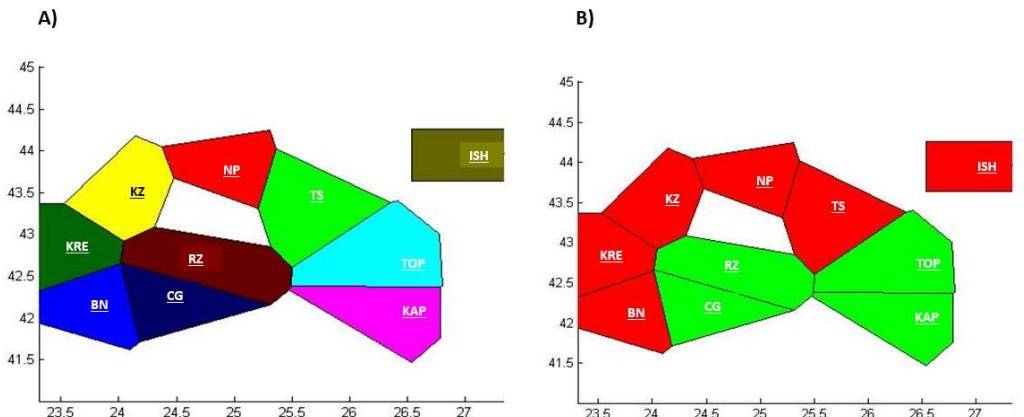

**Figure 3.** Spatial clustering of groups of individuals of EGS constructed in the program BAPS (based on msats data). Same colour represents populations with similar genotypic composition. (**A**): The best model suggests clustering into 10 different populations, which means significant differentiation among studied populations. (**B**): The spatial model with the fixed K-number of populations based on structuring from NJ tree (see Supplementary Materials), which grouped populations into two clusters. Each polygon represents one population corresponding to one line in the Structure analysis and colours of polygons indicate clusters.

The AMOVA test performed for K = 2, i.e., two groups of populations (northern and southern Bulgarian) explained only 5.05% of the genetic variability as from among the groups, while within the groups it was 17%, and the remaining 77.95% were explained by variability within populations ($p < 0.00$).

We found significant genetic differentiation between all populations using pairwise $F_{ST}$ in ENA as a measure of genetic divergence (Table S2, Supplementary Materials). The mean value of $F_{ST} = 0.184$ at the level of significance $p < 0.001$ represents a strong diversification.

We found no significant differentiation between the two groups of population clusters in $F_{ST}$, allelic richness, observed and expected heterozygosities, $F_{IS}$ and corrected relatedness coefficient (Table 1).

No significant IBD was observed for the whole data set (r = 0.051, $p < 0.379$). When we tested partial IBD within the groups, we found significant IBD in the northern Bulgarian group (r = 0.541, $p < 0.038$) and no significant IBD in the southern Bulgarian group (r = 0.162, $p = 0.372$).

## 4. Discussion

The genetic variability of Bulgarian populations is high (the average of $He = 0.64$) with a low level of inbreeding coefficients (the average of $F_{IS} = 0.147$) in comparison with populations on the margin of the species distribution (e.g., in the Czech Republic), where habitat fragmentation is very strong, and where depleted genetic variability (mean value of $He = 0.332$) and high levels of inbreeding (values ranged from 0.553 to 0.907) are also found [48]. This is in accordance with the connectivity models, which reveal that the suitable habitats for the species are still considerably more interconnected in the south [49] and that most of the populations were interconnected until before the recent agricultural intensification and pasture abandonment [21]. Depleted genetic variation in EGS was found on neutral markers as well as on markers of adaptive variation (MHC gene) [22,50]. In Serbia or Hungary the situation is better than in the Czech Republic, and results showed higher genetic diversity [27,51]. For comparison, similar trends were found in *S. suslicus*, where genetic variability was found to range for *He* from 0.37 to 0.69 with mean = 0.54 [6], and all of the genetic diversity indices were significantly more variable in the populations east of Odessa, which are more abundant in comparison with populations from Poland and the western part of Ukraine.

Within Bulgaria, the spatial distribution of EGS populations revealed division into two genetic lineages, i.e., northern and southern Bulgarian (Figure 2). The southern populations, with a centre located in the Trace lowland, along the Maritza and Tundzha rivers, are genetically close to those in the European part of Turkey [22]. This is not surprising given the lack of any physical barriers in that part of the range. However, several clades occur in close proximity here, which is probably related to the fact that this is the area considered the cradle of the species' formation [22], with the oldest fossil records being found nearby in the Yarimburgaz Cave [52].

The northern Bulgarian group is predominantly found in the Danubian plain and north of it in the east of the EGS distribution area. It is located in the extensive lowlands of the east of the country, where the huge pastures are located. It mostly belongs to clade I, which is spread in the whole northern range of the species' distribution [22]. Opposite to what we expected, the Danube river did not show any considerable barrier effect, probably because of changes in the river bed during the Quaternary [53,54]. Further north outside Bulgaria, the big rivers had noticeable barrier effects [27].

The Stara Planina Mountain range almost completely divides the two groups in Bulgaria latitudinally (Figure 4), playing the decisive role of a phytoclimatic barrier [55]. Such a role of mountains with unfavourable habitats in regard to the steppe species has been found across Europe [56]. The northern slopes of Stara Planina and the Pre-Balkan Mountains are characterised by higher precipitation, more days with snow cover and later onset of spring [55]. Southern slopes and adjacent Podbalkan valleys provide warmer and drier conditions, resulting in some differences in the vegetation. For example, *Quercus cerris* and

*Q. franetto* are common in northern Bulgaria, while *Quercus pubescens* and *Carpinus orientalis* dominate in the southern part of the country [57]. The mountain chain is a known distribution barrier for numerous steppe species such as the steppe polecat (*Mustela eversmanni*), common hamster (*Cricetus cricetus*) or Romanian hamster (*Mesocricetus newtoni*) [58]. It also separates the two subspecies of house mouse, *Mus musculus*—*M. musculus musculus* and *M. musculus domesticus* [59]. Thus, it is not surprising that it also separates the two genetic lineages of *S. citellus* in Bulgaria. In this case, the barrier effect is not physical or climatic though, as the species occurs up to the highest parts of the mountain range [21], but is probably due to the dense and vast forests that cover the Pre-Balkan and the northern slopes of Stara Planina [21]. It is also known that the Carpathian Mountains have acted as a barrier for the species' distribution in the northern part of the range [22,53], and it spread to the west of them only in the late Pleistocene. Based on Popova et al. (2019), this dispersal event is the most recent for *S. citellus*, as the species appears in the paleontological record of the Pannonian Basin only in the past 8 Kya [54].

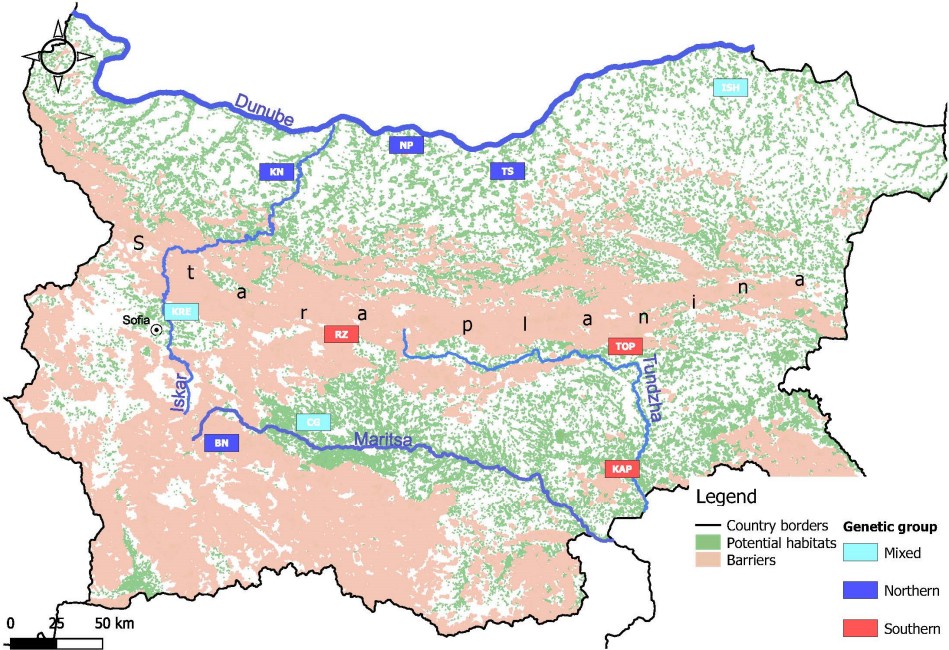

**Figure 4.** Map of the barriers and the potential habitats of the European ground squirrel in Bulgaria (according to [60]). The different genetic groups based on Structure results (Figure 1) are shown with different colours according to the legend.

Several populations in Bulgaria were also assigned to the northern Bulgarian group although they are found south of the Stara Planina Mountains, and two of these are located in higher elevations: BN in the Rila Mountains and KRE on the border of Sofia Valley, one of the Podbalkan Valleys. The situation concerning the Belmeken population in the Rila Mountains is particular. This population inhabits the east Rila Mountains from 1700 to 2500 m a.s.l. and is completely isolated from the lowland populations. The shortest straight-line distance between it and the lowland colonies is more than 19 km, with differences in altitudes of 1.7 km and a forest barrier [61]. The history of its occurrence is unclear. It is possible that the species managed to move through the western part of the Stara Planina where the mountains are cut through by the Iskar River Gorge, because all populations from the northern Bulgarian group that are found south of the Stara Planina Mountains are located in the western area. River valleys are considered as providing suitable corridors for the dispersal of the *Spermophilus* species [27,53].

Based on Popova et al. [53], who proposed a new timescale for the phylogenetic tree of the European ground squirrel compared with Říčanová et al. [22], the separation between the KRE and BN population is about 10,000 years old. Pollen analysis and

radiocarbon chronology testify that the vegetation in the central part of the Rila Mountains 14,000–11,700 years ago was dominated by *Artemisia*, *Chenopodiaceae* and *Poaceae* forming open habitats [62]. Therefore, at that time there was no forest barrier for the spread of the species towards the higher parts of the mountain and it is possible that the split of the genetic lineages occurred afterwards. The same change in vegetation cover, however, also occurred in all the previous glacial–interglacial cycles and is thus compatible with many different datings during the Quaternary.

When we compared the genetic indices between these two groups in Bulgaria, we did not find any differences. The populations in both groups exhibit almost the same variables with high values of allelic richness and expected heterozygosity. Additionally, a recent significant bottleneck was found in only one population (NP), but the concrete reason is unclear, because the species is widespread in the Danube Plain. One possibility could be the high level of coccidian infection in these regions [63], which is suspected to lead to severe population declines [64]. Additionally, when we tested IBD, we did not find a significant correlation, which means that the gene flow is not restricted among all the studied populations cross-country. When we tested partial IBD, there was a significant correlation only for the northern Bulgarian populations, indicating limited migration and gene flow among northern, but not among southern populations.

## 5. Conservation Implication and Role of the EGS in the Ecosystem

The European ground squirrel plays a key role in the ecosystem. Its burrows are inhabited by specific coprophagous beetles or vertebrates such as toads, lizards and snakes, which can also find shelter in the tunnels [65]. EGS represents an important food source for rare and endangered predators such as the steppe polecat (*M. eversmanii*), marbled polecat (*Vormela peregusna*), eastern imperial eagle (*Aquila heliaca*) and the saker falcon (*Falco cherrug*) [66–68]. Thus, the protection of the species is important for the functioning of the whole grassland ecosystem. Unfortunately, it has been severely affected by the historical socio-economic development [28], especially in the northern part of its range [69]. In Bulgaria, being the cradle of the species formation and because of that initially harbouring more genetically diverse populations, it has not been gravely affected until recently. However, the progressive urbanisation, infrastructure construction, intensification and modernisation of the animal husbandry seem to already have serious effects [21]. It is striking to see that the most genetically diverse colony in terms of both allelic richness and overall genetic diversity—TOP—when analysed [22] is now on the brink of extinction after a significant portion of its area has been converted to arable land (30% in 2020, Koshev, Kachamakova—pers. obs.), despite the low farming value of the soil. The colony was once one of the biggest and the most famous in the country for its spectacular concentration of rare raptors. It has also been intensively used as a donor colony for reintroductions and reinforcements (in western Strandzha, Sinite kamani and Kotlenska planina—in Koshev et al. [70]). Almost entirely deprived of legal protection, the colony has collapsed, and it looks like these translocations turn out to be rescue actions. The same pattern of habitat destruction is observed in the whole of south-eastern Bulgaria [71].

This is just a recent example from the field showing how important and urgent it is to acquire and effectively apply the knowledge on population genetics to advocate for appropriate conservation actions. Therefore, the here-presented results should be used to inform the management of the species, including the proper implementation of the National Action Plan that was recently adopted [21]. The proof that the majority of species' genetic diversity is concentrated in the southern part of the range emphasises the responsibility of the national decision-makers. Conservation translocations of individuals should only be carried out after considering the genetic data. The current study shows a high genetic diversity of EGS in Bulgaria and the presence of two phylogenetic lines [22]. Based on these results, it is not recommended to carry out translocations of individuals from different genetic lines, as well as between northern and southern Bulgarian lines. Prompt and substantial measures should be taken to ensure the long-term survival of

the species through maintaining suitable habitats, connectivity, genetic variation and evolutionary potential.

**Supplementary Materials:** The following supporting information can be downloaded at: https://www.mdpi.com/article/10.3390/d15030365/s1, Table S1. Genetic diversity in ten populations of the European ground squirrel based on 12 microsatellite loci.; Table S2. $F_{ST}$ values using ENA (from FreeNA. Chapuis & Estoup 2007) between EGS populations based on msats data.; Figure S1 Neighbour-joining tree from BAPS based on Nei genetic distances divides studied populations into two clusters., i.e., northern and southern groups of populations.

**Author Contributions:** Conception, study design and sample collection, Š.Ř., Y.K. and O.Ř.; statistical analysis, Š.Ř.; review and editing, Š.Ř., Y.K., O.Ř. and M.K. All authors have read and agreed to the published version of the manuscript.

**Funding:** This study was supported by the Grant Agency of the Academy of Science, Czech Republic (KJB601410816), and was partly supported by the Bulgarian Ministry of Education and Science under the National Research Programme "Young scientists and postdoctoral students-2".

**Institutional Review Board Statement:** Not applicable.

**Data Availability Statement:** Not applicable.

**Acknowledgments:** This study was supported by the Grant Agency of the Academy of Science, Czech Republic (KJB601410816), and was partly supported by the Bulgarian Ministry of Education and Science under the National Research Programme "Young scientists and postdoctoral students-2". The anonymous reviewers are acknowledged for their valuable comments on the manuscript.

**Conflicts of Interest:** The authors declare no conflict of interest.

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
