# Peer review of "The European Ground Squirrel’s Genetic Diversity in Its Ancestral Land: Landscape Insights and Conservation Implications"

_diversity, doi:10.3390/d15030365_

Round 1
Reviewer 1 Report
This is an enriched and finer scaled approach of a previous work (Ricanova et al. 2013) exploring the microsatellite dataset from 173 tissue samples of EGS from 10 populations in Bulgaria. The manuscript is generally well written, the data well analyzed (but see below), and well presented. Nevertheless, there are some issues with both the introduction, a part of the methods used, and the discussion.
Although the authors have presented a fully articulated introduction, they have not been very specific in their aims. The general scope of the paper is to investigate the genetic diversity and populations’ structure, but to which end? The authors need to provide some predictions or some hypotheses that need to test by exploring the genetic diversity of the species: Where do they expect intra-population bottlenecks, where do they expect inter-population connections or exclusions, and what does that mean for specific places/colonies and for the whole population of the species in Bulgaria? These predictions need to be explicitly stated in the introduction. And there is no mention whatsoever about phylogeographic analyses, that should be incorporated in the aims, since they occupy a large part of the paper.
I was wondering whether the authors could have used GIS techniques to plot their data on map layers and interconnect genetic diversity or homogeneity with specific recent or past habitat parameters. IN this case, the authors could more readily explain and more thoroughly support any distribution patterns and isolation of populations, such as the major role of forested area on the northern slopes of the Stara Planina.
Figure 2, 3, 4. In these figures the Rozino population appears as homogeneous belonging to the southern population, whereas in previous works (Ricanova et al. 2013) it appears as an admixture of the northern and southern lineage. This discrepancy is not mentioned or commented in the text. It would have been interesting for the authors to explain this fact, as it is important for the implications of the movements of EGS in the area, which is located in the center of the Stara Planina mountain range. It is also strange that the nearby populations (Kremikovtsi, Chernogorovo) also appear as mixed. BTW, Kremikovtsi was also entirely northern in the previous work.
In general, there is too little information about recent events and the presence of bottlenecks in the studied populations. There is too much information on phylogeographic events (where cytochrome b plays a more important role and microsatellite provide finer approaches) but only a few sentences that deals with interconnections or exclusions between populations, where the advantage of microsatellite actually lies. This needs to be rectified, as it does not represent the title of the manuscript. The authors need to decide where they should focus throughout the discussion and probably balance phylogeographic perspectives and recent events that provide more evidence about the status of the populations. This would also help connect with the last section of the paper which discusses about conservation measures.
Author Response
Answers to the comments of Reviewer 1:
Thank you for the comments and corrections. Authors’ answers are given in italic after each paragraph of the review.
This is an enriched and finer scaled approach of a previous work (Ricanova et al. 2013) exploring the microsatellite dataset from 173 tissue samples of EGS from 10 populations in Bulgaria. The manuscript is generally well written, the data well analysed (but see below), and well presented. Nevertheless, there are some issues with both the introduction, a part of the methods used, and the discussion.
Although the authors have presented a fully articulated introduction, they have not been very specific in their aims. The general scope of the paper is to investigate the genetic diversity and populations’ structure, but to which end? The authors need to provide some predictions or some hypotheses that need to test by exploring the genetic diversity of the species: Where do they expect intra-population bottlenecks, where do they expect inter-population connections or exclusions, and what does that mean for specific places/colonies and for the whole population of the species in Bulgaria? These predictions need to be explicitly stated in the introduction. And there is no mention whatsoever about phylogeographic analyses, that should be incorporated in the aims, since they occupy a large part of the paper.
Answer: Thank you for the valuable suggestion. The aim of the study is revised, predictions and hypothesis are included in the end of the Introduction section.
Our aim with this study of genetics is to give to the conservation researchers and practitioners precise and clear criteria in the efforts for preservation of the species’ populations. According to unpublished data in Bulgaria there are several hundred populations (probably over 500) and a certain number of unknown and small colonies located in important stepping stones (Koshev, Kachamakova - unpublished data). Thus, this high number of populations does not allow an initial hypothesis to be made about putative interactions that may be observed between populations. Another factor to be taken into account is the rapid change of habitats after Bulgaria's accession to the EU when accelerated processes of intensification of agriculture and animal husbandry began being stimulated by the EU funds (Koshev 2022). Thus, any conclusions drawn in such an unstable environment will be speculative.
We take our study as a starting point for the conservation of the S. citellus in Bulgaria, and the data from it can be used for:
- Protection of colonies
- Preparation of distribution models and maps and delineation of genetic relationships between populations.
- Planning conservation measures and rescue and environmental protection translocations. These activities are widely implemented at the moment, but the genetic data are not always taken into account.
- For the first time, European ground squirrel’s genetics is being done in Bulgaria. Data from this study can be used for comparison in future studies and determination of the conservation status of the species, that are not possible in the current study.
I was wondering whether the authors could have used GIS techniques to plot their data on map layers and interconnect genetic diversity or homogeneity with specific recent or past habitat parameters. In this case, the authors could more readily explain and more thoroughly support any distribution patterns and isolation of populations, such as the major role of forested area on the northern slopes of the Stara Planina.
Answer: We have replaced Figure 4 with a new one, incorporating GIS models of potential habitats and barrier for species according the official report by Koshev, Popov 2013. Figure 2 and figure 4 illustrate the barrier role of the altitude and forested areas respectively. The populations included in the study are plotted on the maps the the genetic lines being indicated by different colours. Additional clarifications are included in the figures’ captions.
Figure 2, 3, 4. In these figures the Rozino population appears as homogeneous belonging to the southern population, whereas in previous works (Ricanova et al. 2013) it appears as an admixture of the northern and southern lineage. This discrepancy is not mentioned or commented in the text. It would have been interesting for the authors to explain this fact, as it is important for the implications of the movements of EGS in the area, which is located in the center of the Stara Planina mountain range. It is also strange that the nearby populations (Kremikovtsi, Chernogorovo) also appear as mixed. BTW, Kremikovtsi was also entirely northern in the previous work.
Answer: Here, the data were analysed again separately and we are speaking here about northern and southern groups of populations within only in Bulgarian area based only on microsatelite data, not in the whole context. We do not speak about the phylogeographic groups based on mitochodrial DNA as they were mentioned in Ricanova et al 2013. We changed the terms in the article to "Bulgarian southern group" and Bulgarian northern group" for the results in this paper, and use "Northern" and "Southern" groups when referring to the 2013 paper results. Also, results from Structure and BAPS give you different results in terms of the methodology. In Structure you can see individuals but in BAPS information about the whole population, population assignment. In BAPS you will not get a mixed population, the major signal/assignment will be displayed.
In fact, Rozino is not located in the centre of Stara Planina, but in lowlands between two mountains: Stara Planina and Sredna Gora, where the pastures cover big areas and are well interconnected. It is visible on Figure 4. This example illustrates how difficult is to discuss such features in taking into account the complex structure of the relief and habitats (including forests) in Bulgaria as mentioned above.
In general, there is too little information about recent events and the presence of bottlenecks in the studied populations. There is too much information on phylogeographic events (where cytochrome b plays a more important role and microsatellite provide finer approaches) but only a few sentences that deals with interconnections or exclusions between populations, where the advantage of microsatellite actually lies. This needs to be rectified, as it does not represent the title of the manuscript. The authors need to decide where they should focus throughout the discussion and probably balance phylogeographic perspectives and recent events that provide more evidence about the status of the populations. This would also help connect with the last section of the paper which discusses about conservation measures.
Answer: Unfortunately, there are no data about the status of the studied populations during the last centuries. We know only the general land use trends that are related to agriculture intensification after the World War II which became even more pronounced after 1989. In addition, the livestock numbers are gradually declining during the same period. Most of the data on the impact of the contemporary events concern small areas (Koshev 2008, 2009: the Thracian lowland, a mountain region and the Sofia field). It would be too speculative and unreliable to explain the observed genetic features based on such a scarce data. However, some recent data were included in the Discussion.
We think that the analyses of genetic diversity (heterozygosity and other population genetic indices), isolation by distance and/ or bottleneck are sufficient and can give us information about the recent events which can be detectable by microsatellites and which are mentioned in Discussion part.

Reviewer 2 Report
The authors report on the genetics of European ground squirrels in Bulgaria, aiming at a very detailed analysis of several, if not all, genetic traits and variables. This study is original in itself, as it is the first dealing exclusively on the genetics of European ground squirrels in Bulgaria. In my opinion, the authors cover every genetic issue and leave no questions open in this topic.
Overall use of english is quite good, although there are some minor flaws to be corrected and some lacks of clarity to be eliminated. E.g., the sense of lines 66-67, 203-204 and 222-223 is unclear. I added corresponding comments, suggestions and corrections to the manuscript.
On the whole, the manuscript is clear, relevant for the field and presented in a well-structured manner. The cited references are relevant and as recent as possible. The manuscript is scientifically sound and the results are reproducible. The figures and the table are appropriate and properly show the data. The data are interpreted appropriately and consistently throughout the manuscript; statistical analyses seem adequate to me. The conclusions are consistent with the evidence and arguments presented.

Author Response
Answers to the comments of Reviewer 2:
Thank you for the comments and corrections. Authors’ answers are given in italic after each paragraph of the review.
The authors report on the genetics of European ground squirrels in Bulgaria, aiming at a very detailed analysis of several, if not all, genetic traits and variables. This study is original in itself, as it is the first dealing exclusively on the genetics of European ground squirrels in Bulgaria. In my opinion, the authors cover every genetic issue and leave no questions open in this topic.
Overall use of English is quite good, although there are some minor flaws to be corrected and some lacks of clarity to be eliminated. E.g., the sense of lines 66-67, 203-204 and 222-223 is unclear. I added corresponding comments, suggestions and corrections to the manuscript.
Answer: Thank you for noticing this. The sentences are rephrased. Regarding the citation of Peshev et al. 2004, we believe this is the correct way of citation as required by Diversity journal. All other the comments from the PDF file are taken into account.
Round 2
Reviewer 1 Report
The revised version of the manuscript is significantly better than the previous one and the authors have adequately responded to the comments provided. The authors reckognize the limitations of their study in terms of conservation genetics and, based on their analyses, suggest some conservation measures to mitigate the problem of population decline in Bulgaria. I think that the manuscript now merits publication and will be an interesting contribution to the growing body of literature on S. citellus.